# A Bibliometric Visualization Analysis on Vaccine Development of Coronavirus Disease 2019 (COVID-19)

**DOI:** 10.3390/vaccines11020295

**Published:** 2023-01-29

**Authors:** Dequan Zeng, Jie Wang, Bin Xiao, Hao Zhang, Xingming Ma

**Affiliations:** 1School of Health Management, Xihua University, Chengdu 610039, China; 2Health Promotion Center, Xihua University, Chengdu 610039, China; 3School of Food and Biological Engineering, Xihua University, Chengdu 610039, China; 4Department of Nuclear Medicine, Affiliated Hospital of North Sichuan Medical College, Nanchong 637199, China

**Keywords:** SARS-CoV-2, COVID-19, vaccine, bibliometrics, Citespace, VOSviewer

## Abstract

Coronavirus disease 2019 (COVID-19), beginning in December 2019, has spread worldwide, leading to the death of millions. Owing to the absence of definitive treatment, vaccination against COVID-19 emerged as an effective strategy against the spread of the pandemic. Acceptance of the COVID-19 vaccine has advanced considerably, and vaccine-related research has significantly increased over the past three years. This study aimed to evaluate the content and external characteristics of COVID-19 vaccine-related literature for tracking research trends related to the global COVID-19 vaccine with the means of bibliometrics and visualization maps. A total of 18,285 records in 3499 journals were retrieved in the Web of Science Core Collection database and included in the final analysis. China was the first to focus on COVID-19 vaccine research, while European and American countries started late but developed rapidly. The USA and the UK are the top contributors to COVID-19 vaccine development, with the largest number of publications. The University of Washington and Harvard Medical School were the leading institutions, while Krammer, F. from Icahn School of Medicine at Mount Sinai was the author most active and influential to the topic. *The New England Journal of Medicine* had the highest number of citations and the highest TLS, and was the most cited and influential journal in the field of COVID-19 vaccine research. COVID-19 vaccine research topics and hotspots focused on populations’ attitudes towards vaccination, immunity-related information analysis of spike proteins, the effectiveness and side effects of the COVID-19 vaccine, and the public management of epidemic transmission. The findings of this study provide the global status, research hotspots and potential trends in the field of COVID-19 vaccine research, which will assist researchers in mastering the knowledge structure, and evaluating and guiding future developmental directions of COVID-19 vaccine

## 1. Introduction

Coronaviruses are a large family of RNA viruses, with a single-stranded positive sense RNA genome in the inner surrounded with transmembrane glycoproteins (S proteins) in the outer shell, which usually cause infections of multiple organs in both humans and animals [1]. The virus uses these glycoproteins, which anchor themselves to receptors on the host cells they infect, and results in respiratory, gastrointestinal, and neurological illnesses [2]. Seven coronaviruses that infect humans have been identified, including HCoV-229E, HCoV-OC43, HCoV-NL63, HCoV-HKU1, SARS-CoV, MERS-CoV, and SARS-CoV-2 [3]. In the past twenty years, coronaviruses have caused three large-scale worldwide outbreaks and global public health emergencies, which include SARS-CoV, appearing in Guandong, China in 2002 [4], MERS-CoV, discovered in the Middle East in 2012 [5]. SARS-CoV-2 is the most recently discovered, and created a severe worldwide outbreak beginning in 2019.

Coronavirus disease 2019 (COVID-19) is induced by a new type of SARS-CoV-2, which spread worldwide and caused serious adverse effects on social and economic development [6]. This new coronavirus has infected more people and caused many more deaths than its predecessors, with an R_0_ of between 2 and 3.5 [7], and a mortality rate of approximately 2% [1]. In particular, it is thought that infection with SARS-CoV-2 increases the risk of serious illness in individuals. Thus, blocking transmission routes and preventing infection are the best way to reduce the infection and mortality rate of COVID-19. In addition to the fact that personal protective equipment could reduce the risk of exposure to infection, vaccines could significantly reduce the infection rate and economic losses of the COVID-19 [8,9].

There is no proper treatment, and vaccines can play a crucial role in reducing the morbidity and mortality. Vaccines are being developed at a remarkable speed and on an unprecedented scale. To date, there have been 171 COVID-19 vaccines in clinical development and 198 COVID-19 vaccines in preclinical development [10], and the WHO has approved 11 COVID-19 vaccines for the emergency-use listing [11]. The 11 COVID-19 vaccines for emergency use are shown in the Appendix A. More and more vaccines are being approved for marketing and for the evaluation process by the WHO EUL/PQ.

Since the discovery of SARS-CoV-2, the amount of research data and published literature have been increased speedily in an effort to better understand the disease’s various factors, including transmission, pathophysiology, treatment, diagnosis, and vaccine development [12,13]. Currently, research into COVID-19 vaccine-related protection efficacy in humans is booming.

Bibliometric visualization analysis is the statistical evaluation of publications that can both qualitatively and quantitatively describe current status, research hotspots and future trends [14]. In this study, a bibliometric visualization method was used to analyze the content and external characteristics of COVID-19 vaccine-related research literature, including construction of visualization network mapping for co-authorship countries, institutions, active authors, co-cited references and keywords. The primary objective of this article is to summarize a general overview of COVID-19 vaccine research to provide a reference and more possibilities for future research and application.

## 2. Materials and Methods

### 2.1. Study Design

The study attempted to analyze the global research contributions, current status, and research hotspots on the COVID-19 vaccine, and focus on vaccine effectiveness to encourage further research and information dissemination. A bibliometric visualization technique was used to analyze scientific publications on COVID-19 vaccine research in the Web of Science database.

### 2.2. Data Sources

The database Clarivate Analytics Web of Science Core Collection (WoSCC) was used to search and collect studies (https://www.webofscience.com/wos/, accessed on 10 July 2022). The WoSCC database includes A&HCl, CPCI-S, CPCI-SSH, CCR-Expanded, ESCI, IC, SSCI and SCI-Expanded, which is the most commonly used source for bibliometric analysis.

### 2.3. Search Strategy

To identify publications that were relevant to document, the following search phrases were utilized in the field of TOPIC (TS phrase in Web of Science): TS = (Novel Coronavirus OR Coronavirus OR Corona Virus OR 2019-nCoV OR 2019-New Coronavirus OR 2019-Coronavirus OR 2019-CoV OR nCoV-2019 OR nCoV2019 OR New Coronavirus OR Novel Coronavirus 2019 OR Coronavirus 2019 OR Coronavirus-2019 OR COVID-19 OR COVID19 OR COVID-2019 OR Corona Virus Disease-2019 OR Coronavirus Disease 2019 OR Coronavirus Disease-19 OR Coronavirus-2019 OR SARS-CoV-2 OR SARS coronavirus 2 OR SARSCoV2 OR Severe Acute Respiratory Syndrome Coronavirus 2) AND (Vaccine OR Vaccination OR Vaccin*). The timespan was from 1 December 2019 to 10 July 2022, the literature type was article and the language was English. We used Zotero software (https://www.zotero.org/ accessed on 10 July 2022) to screen records and exclude duplicates. The search strategy of this study is shown in the flowchart (Figure 1).

### 2.4. Selection Criteria and Collection

All the documents in the field of COVID-19 vaccine and found through the WoSCC advanced search were included in this study. The following studies were excluded: (1) reviews, editorial materials, letters, case reports, meetings, abstracts, patents and news; (2) duplicate studies; (3) non-English language articles. Following the above criteria, documents were independently identified by two investigators (J.W. and B.X.), and discrepancies were solved by consensus or with a third investigator (H.Z. and X.M.). The following information was recorded from each retrieved article: author, institution, title, abstract, keywords, journal, and publication date.

## 3. Data Analysis

All documents for inclusion were analyzed using CiteSpace software (v.5.8.R3, Philadelphia, PA, USA) and VOSviewer (v.1.6.17, Leiden, The Netherlands).

CiteSpace software [15] was used to analyze information about co-authorship countries, institutions, co-cited references clusters, and keywords of strongest citation bursts. The centrality scores and keywords of citation bursts and heterogeneous networks were detected using CiteSpace software, which can help to visualize the research status, hot spots, and frontiers in a timely manner [16].

The centrality indicator for the nodes of the co-citing network is betweenness centrality. The indicator of centrality is used to assess the literature’s importance and the level of each piece of literature as part of a co-citation [17]. In the CiteSpace, the normalized value of centrality is between 0 and 1, and the nodes with high centrality scores are important documents in a co-cited network [17]. For the keywords of citation bursts, the burst detection algorithm can be suitable to measure sharp increases in interest within a short timespan [18]. According to citation burst terms extracted from bibliographic records, a present research hot spot and frontier is distinguished in CiteSpace software [15,18]. The metrics of mean silhouette (S) and modularity (Q) were used to evaluate the overall structural properties of the network. A silhouette value above 0.7 (S > 0.7) is considered an efficient and convincing cluster [19]. If the value of modularity is more than 0.3 (Q > 0.3), this means that the network is significant [17]. The parameter settings, including time slicing (2019 to 2022, one year per slice), term source (all selected), node type (one at a time), and a selection criterion (g-index = 25 in each slice) were selected while creating a visualization analysis with CiteSpace in the current study [20].

VOSviewer software, developed by van Eck, NJ. and Waltman, L. for viewing bibliometric maps [21], was used to analyze the status of scientific collaborations between authors and journals, and keywords’ co-occurrence in the vaccine research of COVID-19. Two standard weight attributes are applied; these are defined as the “links attribute” and the “total link strength (TLS) attribute” [22]. For a given item, the links and TLS attributes indicate, respectively, the number of links of an item with other items, and the total strength of the links between an item and other items [23]. For example, in the case of co-authorship links between researchers, the TLS attribute shows the total strength of the co-authorship links of a given researcher with other researchers. The following parameter settings were selected for the create-map wizard [23]: type of analysis (co-authorship), counting method (full counting), unit of analysis (authors), minimum number of documents of author (4) and other parameters (default values) for the authors’ co-authorship network visualization map, type of analysis (co-citation), counting method (full counting), unit of analysis (cite source), minimum number of citations of source (22), and other parameters (default values) for the journals’ co-citations network visualization map.

SPSS (IBM SPSS Statistics 19, Inc., Chicago, IL, USA) software was used to analyze the Pearson’s correlation between the total article output, the citations and the total link strength of publications. Correlations were considered to be significant when the *p* value < 0.05.

## 4. Results

### 4.1. Publication Outputs

From 2019 to 2022 (retrieval date 10 July 2022), a total of 18,285 articles published in 3499 journals were included in the current analysis, and are shown in Figure 2 and the Appendix A. The number of papers published in 2021 on the development of COVID-19 vaccines increased rapidly compared to those on the COVID-19 virus epidemic in 2020. Because of COVID-19 virus mutations and the widespread outbreak all over the world, scientists focus on the development of safe and effective vaccines to control the COVID-19 virus epidemic.

According to the topic words of COVID-19 vaccine names and types, there were a total of 1914 articles published on 11 WHO-approved COVID-19 vaccines among 18,285 included records (10.5%), which were showed in the Appendix A. Among the four classifications of the COVID-19 vaccine, the vaccine research with the most interest concerned messenger-RNA (mRNA) vaccines, with 1210 publication documents (63.2%); this was followed by recombinant adenovirus vector vaccines, with 397 publication documents (20.7%), inactivated COVID-19 vaccines, with 231 publication documents (12.1%), and recombinant S proteins subunit vaccines, with 76 publication documents (4.0%).

### 4.2. Main Publication of Countries, Institutions, Authors and Journals

The top 10 countries that contributed most to publications in COVID-19 vaccine research are listed in Table 1, showing their productivity and scientific influence. Among the total countries, six countries produced more than 1000 documents. The most productive country was the USA, with 5711 (31.2%) publication documents, followed by the UK (*n* = 2020, 11.05%), China (*n* = 1819, 9.95%), Italy (*n* = 1253, 6.85%), and India (*n* = 1193, 6.52%). Among the ten most productive countries, the top 3 countries with the highest centrality ranking were the UK (0.23), Germany (0.18) and the USA (0.05).

The top 10 most prolific institutions concerning COVID-19 vaccine research are described in Table 2. Of the total institutions, 40 institutions published more than 100 documents, and 3 institutions published more than 300 documents about COVID-19 vaccine research. The most prolific institution was the University of Washington with 340 (1.9%) publication documents, followed by the University of Oxford (*n* = 337, 1.8%), and Harvard Medical School (*n* = 309, 1.7%). The Chinese Academy of Sciences ranked fifth, with at least 210 documents on COVID-19 vaccine research. Among the ten most productive institutions, the top 3 institutions with the highest centrality ranking were Harvard Medical School (0.38), Tel Aviv University (0.27) and the University of Washington (0.22).

The most prolific authors in COVID-19 vaccine research were Krammer, F. (*n* = 32, Citations = 3217) and Yuen, KY. (*n* = 32, Citations = 3142), as shown in Table 3. Of the total authors, 236 authors published at least 10 documents about COVID-19 and vaccine research. Among the ten most prolific authors, the top 3 authors with the highest TLS ranking were Yuen, KY. (271), To, KK. (234) and Krammer, F. (198).

Among the total journals, four journals published more than 1000 documents. The most prolific journals on the COVID-19 vaccine were *Vaccines* (*n* = 965; 5.3%), followed by *Vaccine* (*n* = 365; 1.9%), *Plos One* (*n* = 327; 1.8%), the *International Journal of Environmental Research and Public Health* (*n* = 320; 1.7%), and *Human Vaccines Immunotherapeutics* (*n* = 294; 1.6%), as described in Table 4.

### 4.3. Visualization Map of Co-Authorship Countries and Institutions

Figure 3A,B show a linking map of co-authorship countries and institutions. A complete picture of the academic performance of the leading countries and institutions is described in Figure 3. The size of the circles represent the number of articles published by the country or institution, and the larger the circles, the higher the co-authorship of the countries or institutions. The thick lines between two countries or institutions suggests more collaboration between those two countries or institutions. There were 188 nodes and 223 links in the countries network map, and 697 nodes and 749 links in the institutions network map. The low density of the country-collaboration and institution-collaboration network map suggests that most countries and institutions were fragmented, and lacked consistent and extensive cooperation.

The USA was the most productive country (*n* = 5711). It was one of the leading countries with the largest network of international cooperation in COVID-19 vaccine research. The UK (*n* = 2020) ranked second, and third was China (*n* = 1819), in the number of published documents. According to publication and centrality (Table 1), the USA, UK, and Germany were the major research powers in the COVID-19 vaccine field. Although developing country such as China and India ranked in the top five in the number of published papers, there was still a certain gap between the centrality and the developed country.

Among these institutions, the University of Washington (*n* = 261, centrality = 0.38), the University of Oxford (*n* = 377, centrality = 0.34), and Harvard Medical School (*n* = 363, centrality = 0.27) published more than 100 documents. The publication and centrality statistics show that advanced techniques and core competences for COVID-19 vaccine development were in those developed countries and institutions.

### 4.4. Visualization Map of Active Authors and Journals

In COVID-19 vaccine research, the co-authorship network of the most active authors is shown in Figure 4. More than 99,181 authors co-authored publications on COVID-19 vaccine research. The authors who collaborated on at least four documents are shown in Figure 4. The nodes in Figure 4 represent the authors, and the links in Figure 4 represent the number of co-authorships formed between different authors. The size of nodes represents the number of documents and is positively correlated with the connection strength of the authors, and authors with the same color in the view belong to the same clustered cooperation network. As shown in Figure 4, eight major clusters of authors can be distinguished in the co-authorship network. Among these authors in clusters of collaborators’ networks, Krammer, F. (32 of documents) was the most active researcher whose many documents have been published on COVID-19 vaccine research.

Figure 5 shows the co-authorship network of the most active co-cited journals in the vaccine research of COVID-19. The top 2980 journals, in which the number of co-citations was more than 22, are shown in Figure 5 and Appendix A. The size of the nodes represents the citation frequency of journals, and journals with the same color in the view belong to the same clusters in the cooperation network. *The New England Journal of Medicine* was the leading source, with the highest TLS (966,309) and citations (23,918), followed by *Nature* (TLS = 920,635, citations = 17,873), *Science* (TLS = 775,414, citations = 14,850), *The Lancet* (TLS = 701,295, citations = 16,443), *Cell* (TLS = 602,806, citations = 10,514), and *Vaccine* (TLS = 522,969, citations = 13,160). The top 20 active co-citation cited sources visualization map is presented in Figure 5.

The results of correlation analysis with Pearson’s correlation indicated that a positive correlation was found between total articles output, citations and TLS of journals on the research of the COVID-19 vaccine (*p* < 0.05). A significantly strong correlation between the TLS and citations of journals with the correlation coefficient R = 0.98 (*p* < 0.01), a weak correlation between the number of articles and citations of journals (R = 0.33 and *p* < 0.01), and a negligible correlation between the number of articles and the TLS of journals (R = 0.27 and *p* < 0.01) were found in the cooperation network on the research of COVID-19 vaccine.

### 4.5. Visualization Map of Co-Cited References Cluster

The cluster analysis of document co-citation could help us to understand their interconnection and the emerging trends in this field. According to the selection criterion of the g-index in each slice, a minimum citation frequency of 10 was set to screen, and then 22 co-citation clusters were generated; the largest 19 clusters are shown in Figure 6. All clusters were labeled by keywords extracted from the references.

Of the total clusters, four clusters had more than 120 members and eight clusters had more than 100 members carrying out COVID-19 vaccine research. The largest cluster (#0) had 163 members, with a silhouette value of 0.99, which was labeled as vaccine hesitancy. The second largest cluster (#1) had 162 members, with a silhouette value of 0.88, which was labeled as spike protein. The third largest cluster (#2) has 161 members, with a silhouette value of 0.88, which was labeled as molecular docking. The fourth largest cluster (#3) has 129 members, with a silhouette value of 0.94, which was labeled as vaccine. Attitudes towards vaccination, spike protein, the effectiveness and side effects of COVID-19 vaccines were the three main related research topics.

In the co-citation clusters analysis, the top 10 highest-citation documents ranked by citation counts are summarized in Table 5. The first ranked item by citation counts was Polack, FP. (2020) in Cluster #3, with a citation count of 2521. The second one was Baden, LR. (2021) in Cluster #3, with a citation count of 1610. The third was Zhu, N. (2020) in Cluster #2, with a citation count of 995. The fourth was Voysey, M. (2021) in Cluster #3, with a citation count of 958. The fifth was Hoffmann, M. (2020) in Cluster #1, with a citation count of 940. 

### 4.6. Visualization Map of Co-Occurrence Keywords and Burst Keywords 

In this study, the VOSviewer was used to analyze the co-occurrence of keywords for presenting the distribution characteristics of COVID-19 vaccine research hotspots, and then the keywords were analyzed by clustering. In Figure 7, the size of the nodes represented the frequency of the keywords, and the larger of the nodes suggests the main research content. The connecting line indicates the co-occurrence relationship between keywords, according to which different research clusters were formed, and the keyword co-occurrence mapping in the field of COVID-19 vaccine was drawn (Figure 7). As shown in Figure 7, the term “COVID-19” had the largest node and the highest frequency, followed by “SARS-CoV-2”, and “vaccine”, respectively. Because COVID-19 constituted the core concepts in the keyword co-occurrence mapping, other high-frequency keywords based on co-occurrence relationships are presented the main research clusters; these can be used to analyze the main research hotspots and frontiers in this field. This knowledge about epidemiology, biology characterization, pathogenicity, mutation and the serological testing of SARS-CoV-2 is an important stepping stone to the successful development of vaccines and treatments.

Afterwards, the keywords with the strongest citation bursts were also detected and analyzed with CiteSpace (Figure 8). The top five strongest burst keywords included coronavirus (63.69), respiratory syndrome coronavirus (38.88), SARS coronavirus (31.94), spike protein (22.64), and acute respiratory syndrome (16.47), which formed the greatest class of attention on the structure, aetiology, and entrance mechanism research of SARS-CoV-2 during the period between 2019 and 2020. This knowledge forms important parameters for the development of effective vaccines and medicines. In addition, from the second-class strongest citation burst keywords, such as spike protein, glycoprotein, recombination, docking and antibody, it can be inferred that previous publications focus on protein vaccine and protective efficiency against COVID-19.

## 5. Discussion

COVID-19 is a global pandemic caused by severe acute respiratory syndrome coronavirus 2 (SARS-CoV-2). We are all in the midst of the worldwide epidemic and in the ongoing battle against this infection disease. After the emergence of COVID-19, information and knowledge on it have been growing rapidly. Scientific data and research play a very important role in the prevention and control of COVID-19 outbreaks and epidemics. Unlike previous bibliometric analysis on COVID-19 topics [34,35], the latest literature data were focused on the vaccine research of COVID-19. In the present study, more than 18,285 documents retrieved from the WoSCC database have been published on COVID-19 vaccine-related research. Among these publications, 2400 papers were published in 2020 and 10,290 were published in 2021.

As of 1 August 2022, 67.4% of the world population has received at least one dose of a COVID-19 vaccine, and 24.2% of people are fully boosted with a COVID-19 vaccine [36,37]. A total of 12.43 billion doses have been administered globally, and 6.6 million are now administered each day [36]. However, only 20.7% of people in low-income countries have received at least one dose of a COVID-19 vaccine [36]. Vaccination programs remain very important for populations in those low-income countries. Particularly, with the persistent mutation and the ongoing pandemic of SARS-CoV-2 globally, there is an urgent need for a much more effective and safe COVID-19 vaccine.

At the beginning of the COVID-19 outbreak, Chinese scientists shared the complete genome sequence information of 2019-nCoV (GenBank: MN908947) with other countries, for the development of new treatment and vaccines [38]. Since then, initiatives announced by the WHO have advanced research into COVID-19 vaccines. The bibliometric analysis in this study indicates that the USA, the UK, China, Italy, India, Germany and Canada have made outstanding contributions to this important field. Particularly, the USA, the UK, and Germany were the major research powers in the COVID-19 vaccine field. Although the University of Washington and the University of Oxford are the most prolific institutions in terms of published articles, Harvard Medical School and Tel Aviv University are the most active institutions based on centrality. In the study, a low degree of node centrality and a low density of country collaboration and institution collaboration networks was observed in most countries and institutions, which indicated less international and inter-institutional collaboration. COVID-19 vaccine development was uneven among different countries or regions, and the academic exchange was insufficient; this might negatively impact the development of the field. With the spread of the pandemic, research related to COVID-19 vaccine has attracted a lot of attention worldwide [39]. Therefore, we strongly suggest that countries and their institutions need to break down academic barriers and expand cooperation and exchange to jointly promote the progress of COVID-19 vaccine research.

Although the top ten most prolific medical journals, such as *Vaccines*, *Vaccine*, *Plos One,* etc. have published over 18.0% of the relevant articles of the total publications, they had weak influence on the guidance of COVID-19 vaccine research directions for researchers. Meanwhile, the highest-citation and highest-TLS medical journals, such as the *New England Journal of Medicine*, *Nature*, *Science*, *Lancet,* and *Cell*, published SARS-CoV-2 vaccine-related studies which identified them as the most valuable journals on COVID-19 vaccine research in this field. Of interest, most of these publications have focused on the relationship between SARS-CoV-2 biology and COVID-19 vaccine efficacy. This study also highlighted the contributions of influential researchers who could provide guidance for further research directions. It is worth noting that Krammer, F. was the most influential scholar in this field, with the highest scores in both the number of citations (3217) and documents (32), indicating an outstanding contribution to the field of COVID-19 vaccine development. His most cited study (cited 870 times) was published in the journal *Nature* in 2020 [40]. This article reviewed the available genetic sequence of the virus and discussed all the challenges of vaccine development against SARS-CoV-2, which provided a countermeasure against SARS-CoV-2 to promote the development of vaccine. Then, he focused on the targets of T cell responses to SARS-CoV-2 coronavirus; this is important for vaccine development, for interpreting COVID-19 pathogenesis, and for calibration of pandemic control measures [41].

The cluster analysis of references is one of the most significant indicators of bibliometrics, and is generally used to mine their interconnection and the emerging trends in the field. In this cluster analysis of document co-citation, there are 19 high-quality clusters with silhouette value of 0.93 and modularity value of 0.77, which shows that there was homogeneity and that the clusters were significant in the network. The node within the clusters had strong links, and the link between the clusters was powerful.

The largest cluster (#0), labeled ‘COVID-19 vaccine hesitancy’ by keywords, represented population attitudes towards vaccination, which is one of the research hotspots on COVID-19 vaccine. A recent systemic review on the general populations of 33 countries revealed a variation in line with income level and geographical region [42]. There is low acceptance in low-income countries (23.6–28.4%), moderate to half acceptance in European countries (53.7–56.3%), and high acceptance in eastern Asian countries (91.3–94.3%) [43]. Developing effective and safe vaccines is no longer enough to combat a communicable disease due to growth in vaccine hesitancy. Because of people’s general uncertainty towards vaccines and their unwillingness to take vaccines, vaccine hesitancy has been listed among the most pressing issues facing public health [44].

Various factors affecting vaccination rates such as sex, educational level, race, region, psychological health and trust in the government have been reported to influence the population response to vaccination [45,46]. Among those factors, education level is an essential factor influencing the formation of a knowledge gap [47]. Those with higher education levels have more knowledge about COVID-19 disease and COVID-19 vaccination than those with lower education levels. The existence of a knowledge gap around SARS-CoV-2 and COVID-19 vaccination needs more attention during the COVID-19 pandemic [48]. Thus, the government and public health authorities should make multi-pronged efforts and take the necessary steps to increase vaccination acceptance and positive attitudes towards the vaccine.

The second (#1) and third (#2) largest cluster labeled, respectively, by keywords of ‘molecular docking’ and ‘spike protein’, suggested that the spike protein of SARS-CoV-2 is the most preferred target for making vaccines or therapeutics against coronaviruses. Overwhelming attention has been paid to the S protein of the virus. The S protein of SARS-CoV-2 plays the most critical role in viral attachment, fusion, and entry into the target cells [49]. Most SARS-CoV-2 S proteins contain six substitutions in the receptor-binding domain [50], and use angiotensin-converting enzyme 2 (ACE2) as a dominant receptor for cellular entry [21].

The fourth largest cluster (#3), labeled ‘COVID-19 vaccine’ by keywords, indicated that the safety and effectiveness of vaccination is another important focus hotspot in the COVID-19 vaccine research field. Among the top ten most cited publications, three papers in this cluster, which reported the results of safety and effectiveness of three vaccines, the ChAdOx1 nCoV-19 vaccine (AZD1222), the BNT162b2 mRNA vaccine, and the mRNA-1273 vaccine, yielded more than 4760 citations [17,18,20]. In a recent systematic review and meta-analysis, outcomes suggest that COVID-19 vaccines are effective in reducing the incidence of SARS-CoV-2 infection and have negligible adverse effects; this confirmed the effectiveness and safety of the COVID-19 vaccine after vaccination [51,52]. However, these vaccines achieved about 62–94% effectiveness against original SARS-CoV2 in clinical trials [20]. On the other hand, the genetic mutants of SARS-CoV2 and the rapid transmissibility of variants such as alpha, beta, gamma, delta, etc. have worsened the situation [20]. Variants such as delta have a higher chance of evading the immune response induced by COVID 19 post-vaccination, which could potentially lead to a drop in the protective efficacy of the vaccination in individuals [20,53]. Therefore, developing safe and effective novel COVID 19 vaccines and increasing the mass vaccination globally will surely increase the protection level against a wide variety of coronaviruses, and will make it harder for the infection to spread further in future.

Furthermore, in the co-occurrence keywords analysis, the word ‘COVID-19’, which constituted the core relationship, was strongly linked to keywords such as “SARS-CoV-2”, “vaccine”, “vaccination”, “epidemiology”, “transmission”, “infection”, “variant” and “serology” in the co-occurrence clusters map. According to the strongest citation bursts, keywords such as “coronavirus”, “spike protein”, “MERS CoV”, “replication”, “glycoprotein”, “receptor”, “convalescent plasma” were the most strongly associated with COVID-19 vaccine research hotspots. Altogether, using the co-occurrence and burst of keywords, we may predict the hotspots and emerging trends of COVID-19 vaccine research in four areas, as follows: population attitudes towards vaccination, immunity-related information analysis of structural proteins, the effectiveness and side effects of the vaccine, and the public management of epidemic transmission.

There are still several limitations on this research. First, we only chose to include literature in the English language from the WoSCC database, which may lead to the omission of related articles in languages such as Chinese, French or other languages. Second, only the WoSCC database was chosen as a data source to search the COVID-19 vaccine research literature, not including the existing Embase, Medline, or other databases; this may cause a small variation in the number of citations in the results [54]. Third, the chosen document type was limited to article, so the current study did not include publications other than article-type publications, such as reviews, editorial materials, case reports, etc. Fourth, considering that COVID-19 vaccine research continues to progress, this bibliometrics analysis is limited from 1 December 2019 to 10 July 2022, and future studies should add the latest documents to update the development. Fifth, the number of citations for the older articles may be higher than that of the latest articles, but the older research may not keep up with the latest research hotspots [55]. Sixth, one of the reasons for high citation numbers may be authors’ self-citation [56]; this visualization analysis failed to eliminate self-citation, and thus, further research is needed to analyze the frequency of self citation and its impact on the published article. In addition, a total of ten retracted articles (0.05%) were found and not excluded from this visualization analysis, as shown in the Appendix A. In spite of these limitations, bibliometrics analysis is still a valuable means to quantitatively provide a comprehensive overview of the literature to some extent.

## 6. Conclusions

A total of 18,285 records in 3499 journals were retrieved from the Web of Science Core Collection database and included in the final analysis. Over time, the number of articles on COVID-19 vaccine research has increased. Although the highest number of publications were from the USA followed by UK and China, the USA, UK, and Germany were the top contributors to the COVID-19 vaccine research field. The University of Washington and Harvard Medical School were the leading institutions, while Krammer, F. from the Icahn School of Medicine at Mount Sinai was the most active and influential author on the topic. *The New England Journal of Medicine*, with the highest citation and TLS, was the most cited and influential journal in the field of COVID-19 vaccine research. The vaccine research hotspots of COVID-19 focused on populations’ attitudes towards vaccination, immunity-related information analysis of spike proteins, the effectiveness and side effects of the COVID-19 vaccine, and the public management of epidemic transmission. The findings of this study provide the global status, research hotspots and potential trends in the field of COVID-19 vaccine research, which may assist researchers in mastering the knowledge structure, and evaluating and guiding the future development trends of global COVID-19 vaccine research.

## Figures and Tables

**Figure 1 vaccines-11-00295-f001:**
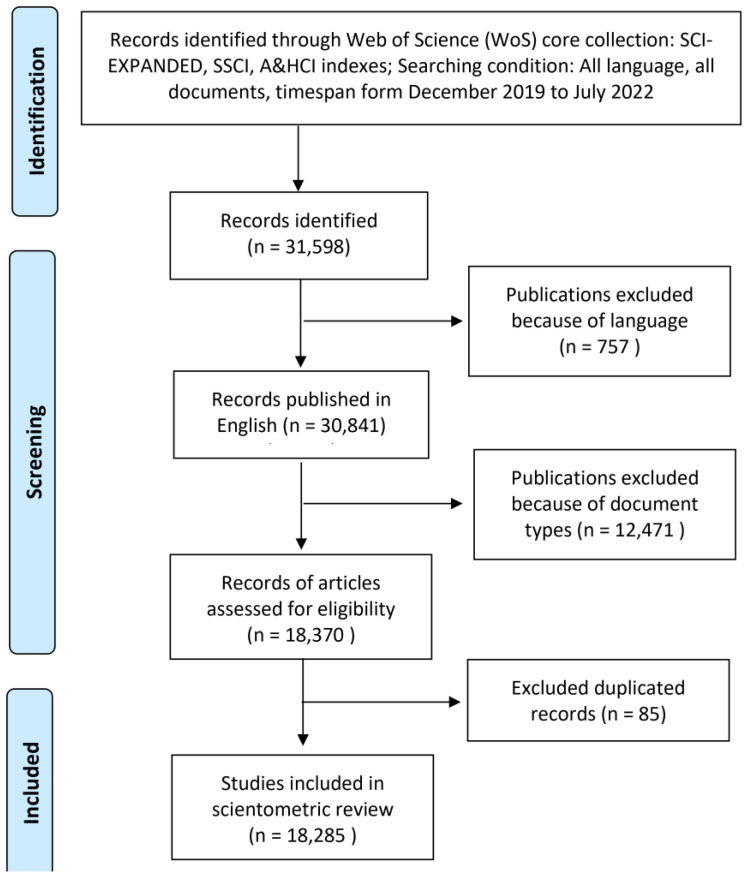
PRISMA flow diagram on COVID-19 vaccine research.

**Figure 2 vaccines-11-00295-f002:**
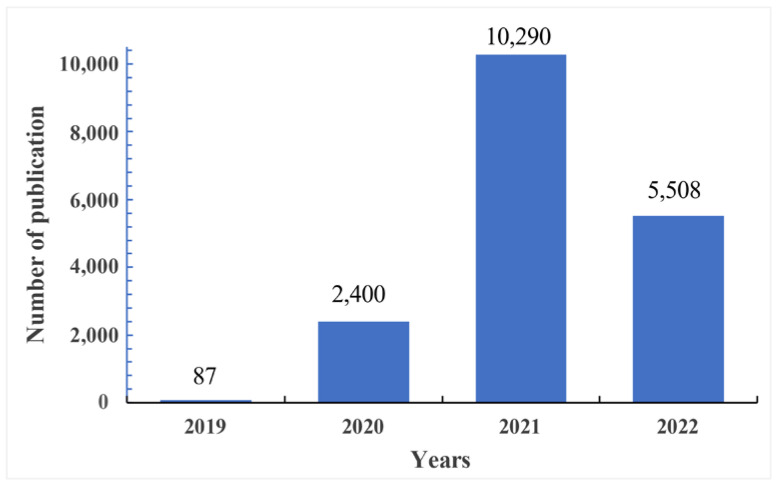
The annual distribution of in COVID-19 vaccine research.

**Figure 3 vaccines-11-00295-f003:**
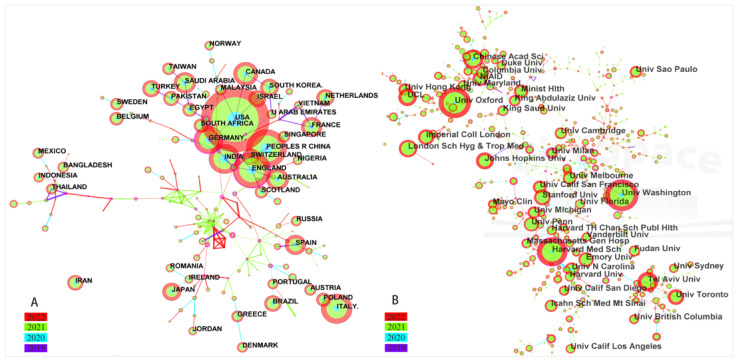
The countries’ (**A**) and institutions’ (**B**) co-authorship network visualization map in COVID-19 vaccine research. The CiteSpace software was used to analyze information about co-authorship countries and institutions. The selection criterion was that the g-index was 25 in each slice.

**Figure 4 vaccines-11-00295-f004:**
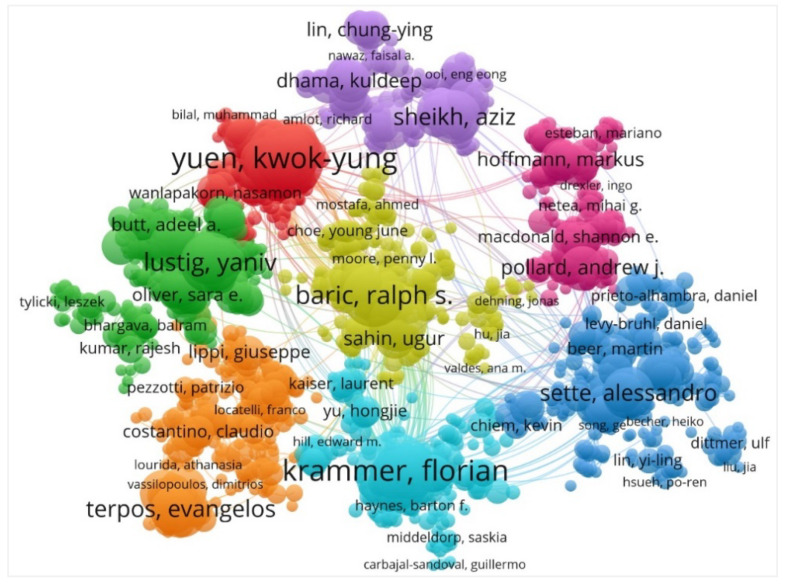
The authors’ co-authorship network visualization map in COVID-19 vaccine research. The authors’ co-authorship network visualization map was produced with VOSviewer, and the top 3300 authors who collaborated on at least four documents are shown in figure. The size of the nodes represents the number of documents, and authors with the same color in the view belong to the same clustered cooperation network.

**Figure 5 vaccines-11-00295-f005:**
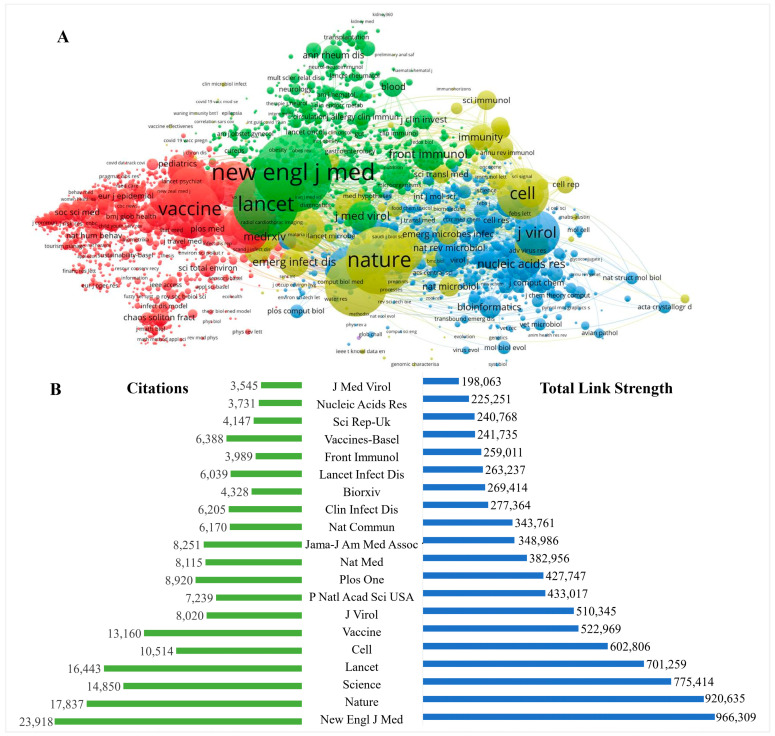
The top 20 active journals and co-citation cited sources visualization map within COVID-19 vaccine research, by VOSviewer analysis. (**A**) network visualization map of top 2980 cited sources (co-citations more than 22); (**B**) the top 20 highest TLS journals. The size of the nodes represents the citation frequency of journals, and journals with the same color in the view belong to the same clusters in the cooperation network.

**Figure 6 vaccines-11-00295-f006:**
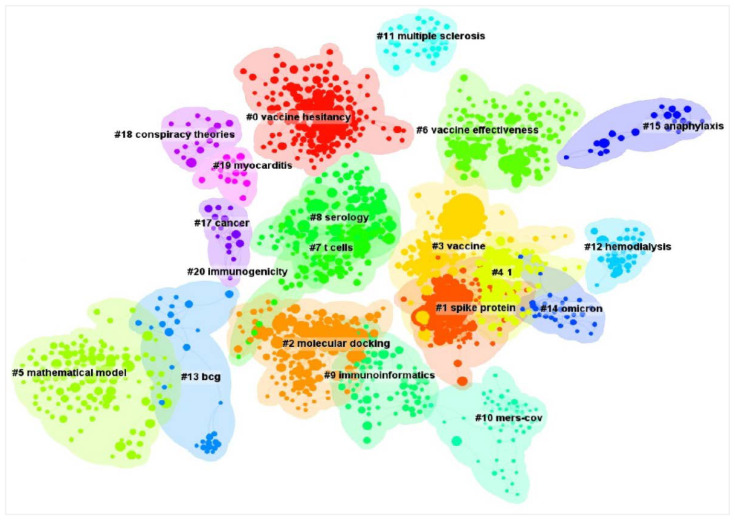
The reference co-authorship clusters visualization map in COVID-19 vaccine research, by CiteSpace analysis. All clusters were labeled by keywords extracted from the references.

**Figure 7 vaccines-11-00295-f007:**
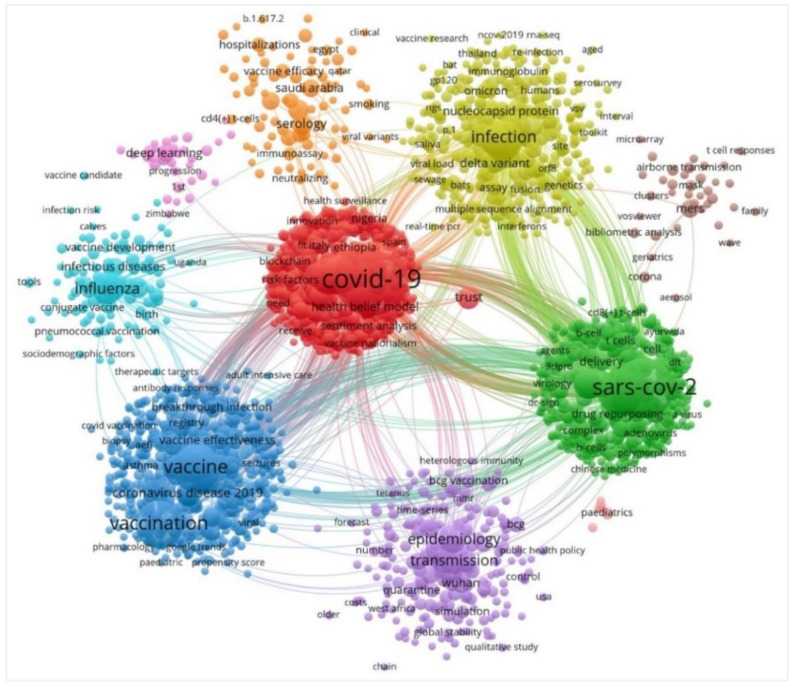
Keywords’ co-occurrence clusters visualization map in COVID-19 vaccine research, by VOSviewer analysis. The size of the nodes represents the occurrences of the keywords, and keywords with the same color in the view belong to the same clusters in the cooperation network.

**Figure 8 vaccines-11-00295-f008:**
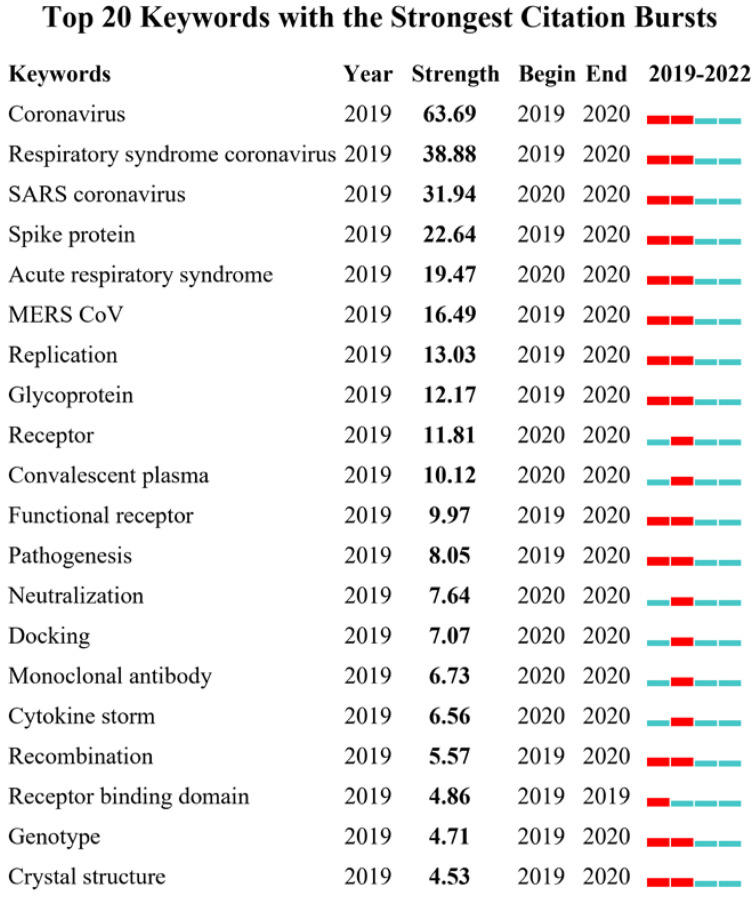
Keywords with the strongest citation bursts in publications on COVID-19 vaccine research, by CiteSpace analysis. The red lines in the diagram are heat bars, representing the time period with the strongest citation bursts.

**Table 1 vaccines-11-00295-t001:** Top 10 most productive countries in COVID-19 vaccine research.

Ranking	Countries	Records	Percentage (%)	Centrality
1	USA	5711	31.23	0.05
2	UK	2020	11.05	0.23
3	China	1819	9.95	0.02
4	Italy	1253	6.85	0.01
5	India	1193	6.52	0.01
6	Germany	985	5.39	0.18
7	Canada	773	4.23	0.03
8	Australia	652	3.57	0.01
9	Saudi Arabia	583	3.19	0.02
10	France	567	3.10	0.02

**Table 2 vaccines-11-00295-t002:** Top 10 most prolific institutions on COVID-19 vaccine research.

Ranking	Institutions	Records	Percentage (%)	Centrality
1	University of Washington	340	1.86	0.22
2	University of Oxford	337	1.84	0.02
3	Harvard Medical School	309	1.69	0.38
4	Tel Aviv University	214	1.17	0.27
5	Chinese Academy of Sciences	210	1.15	0.05
6	University College London (UCL)	192	1.05	0.05
7	The London School of Hygiene & Tropical Medicine	191	1.04	0.02
8	Johns Hopkins University	184	1.01	0.01
9	Stanford University	174	0.95	0.01
10	University of Toronto	171	0.94	0.04

**Table 3 vaccines-11-00295-t003:** Top 10 most prolific authors on COVID-19 vaccine research.

Ranking	Authors	Records	Citations	TLS
1	Krammer F	32	3217	198
2	Yuen KY	32	3142	271
3	To KK	25	3014	234
4	Baric RS	25	954	163
5	Lustig Y	24	875	167
6	Shi PY	23	3071	154
7	Terpos E	22	179	166
8	Diamond MS	21	1634	166
9	Sheikh A	21	235	125
10	Sette A	20	3704	109

(TLS = total link strength).

**Table 4 vaccines-11-00295-t004:** Top 10 most prolific journals on COVID-19 vaccine research.

Ranking	Journals	Records	Percentage (%)	IF
1	*Vaccines*	965	5.3	4.96
2	*Vaccine*	365	1.9	4.16
3	*Plos One*	327	1.8	3.75
4	*International Journal of Environmental Research and Public Health*	320	1.7	4.61
5	*Human Vaccines Immunotherapeutics*	294	1.6	4.52
6	*Frontiers in Immunology*	276	1.5	8.78
7	*Scientific Reports*	270	1.4	4.99
8	*Viruses Basel*	225	1.2	5.82
9	*Frontiers in Public Health*	169	0.9	6.41
10	*Clinical Infectious Diseases*	157	0.8	20.99

**Table 5 vaccines-11-00295-t005:** Top 10 highest-citation documents on COVID-19 vaccine research.

Ranking	Citation Counts	References and Research Topics	Cluster ID
1	2521	[24] Polack et al. evaluated the safety and efficacy of the BNT162b2 mRNA COVID-19 vaccine with a multinational, placebo-controlled, observer-blinded trial which was published in the *Journal of N Engl J Med* in 2020.	3 (#3)
2	1610	[25] Baden et al. conducted a phase 3 randomized controlled trial and investigated the efficacy and safety of the mRNA-1273 SARS-CoV-2 vaccine, which published in the *Journal of N Engl J Med* in 2021.	3 (#3)
3	995	[26] Zhu et al. isolated a novel coronavirus from patients in China, named 2019-nCoV, which is the seventh member of the family of coronaviruses that infect humans and published in the *Journal of N Engl J Med* in 2020.	2 (#2)
4	958	[27] Voysey et al. evaluated the safety and efficacy of the ChAdOx1 nCoV-19 vaccine (AZD1222) against SARS-CoV-2 with an interim analysis of four randomised controlled trials in Brazil, South Africa, and the UK, which was published in *The Lancet* in 2021.	3 (#3)
5	940	[28] Hoffmann et al. demonstrated that SARS-CoV-2 uses the SARS-CoV receptor ACE2 for entry and the serine protease TMPRSS2 for S protein priming, which is a potential target for antiviral intervention, and published their report in the journal *Cell* in 2020.	1 (#1)
6	893	[29] Walls et al. determined cryo-EM structures of the SARS-CoV-2 S ectodomain trimer, providing a blueprint for the design of vaccines and inhibitors of viral entry, and published in the journal *Cell* in 2020.	1 (#1)
7	852	[30] Wrapp et al. determined a 3.5-angstrom-resolution cryo-electron microscopy structure of the 2019-nCoV S trimer in the prefusion conformation to facilitate medical countermeasure development, and published in the journal *Science* in 2020.	1 (#1)
8	755	[31] Huang et al. reported the epidemiological, clinical, laboratory, and radiological characteristics and treatment and clinical outcomes of these patients infected with 2019 novel coronavirus in Wuhan, and published in *The Lancet* in 2020.	2 (#2)
9	638	[32] Lazarus et al. surveyed 13,426 people in 19 countries to determine the potential acceptance rates and factors influencing acceptance of a COVID-19 vaccine, and published in the *Journal of Natural Medicines* in 2021.	0 (#0)
10	614	[33] Zhou et al. confirmed that 2019-nCoV is 96% identical at the whole-genome level to a bat coronavirus, and using the same cell entry receptor-angiotensin converting enzyme II (ACE2)-as SARS-CoV, and published in the journal *Nature* in 2020.	1 (#1)

## Data Availability

The original contributions presented in the study are included in the article and its Appendix A; further inquiries can be directed to the corresponding authors.

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
