# Peer review of "A Bibliometric Visualization Analysis on Vaccine Development of Coronavirus Disease 2019 (COVID-19)"

_vaccines, 2023, doi:10.3390/vaccines11020295_

Round 1

Reviewer 1 Report

This article presents a bibliometric analysis of articles published in the English language on the subject of COVID-19 vaccination. Since I do not have a medical or clinical background, I will limit my comments to the bibliometric content of the article. I assume that at least one of the other reviewers has a background in vaccination and is capable of reviewing those portions of the paper. If this is not the case then the paper should not proceed to publication until this has been rectified.

The authors have taken great pains to acquire a large dataset and they outline their data collection processes thoroughly, allowing others to replicate the data collection, but the analysis is rather superficial. The methodological sections focussing on the analysis require much more detail, and the conclusions are largely just descriptive of the dataset and lack depth.

Data analysis:

There is very little detail about some crucial methodological points. What is "centrality"? What are "citation bursts"?  Simply referring to the article that introduces the CiteSpace software is not enough to make these concepts meaningful for the readership of Vaccines, who will largely be unfamiliar with bibliometrics. The concepts of "link strength" and "silhouette value" which occur regularly in the 'Results' section are never explained.

Results:

This section describes in some detail the surface characteristics of the dataset, but fails to dig under that surface. Thanks to the severity of the pandemic, the COVID-19 scholarly literature went from being non-existent to totally overwhelming within the space of a few short months. As the authors note, 18,285 articles were published in the relatively short period under discussion here, and - crucially - they were published in 3,499 separate journals. Take a look at Figure 5A and tell me whether this is a sensible way to organise the scholarly literature? No group of researchers, even those in prestigious universities, with large library budgets, in rich countries, can be expected to have access to three and a half thousand medical journals. As the pandemic wore on, several academic publishers rushed to make many of these publications available open access, but only because it was politically expedient for them to do so and avoid the accusation that they are holding the world's accumulated knowledge to ransom. Looking at the data presented here, I see the traditional, subscription-journal-based scholarly publishing system reaching (and exceeding) its breaking point. The authors do not go further than simply describing the characteristics of the data.

Similarly, no comparison is made of Table 4 (which shows the most "prolific" journals) and Figure 5B (which shows the journals having the most citation impact). Would a more thorough analysis show a negative correlation between "productivity" and impact?

The section on high[ly]-frequent keywords is of very little interest and could be deleted to make room for more methodological detail. The fact that a large number of papers on COVID-19 vaccination use keywords that are all variations on "COVID-19" and "vaccine" is unsurprising. It needs to be better justified if it is to be included.

The parameters employed in VOSviewer to achieve the clustering shown in Figures 4 and 5 is not made explicit in the 'data analysis' section.

Altmetrics as a concept are not properly introduced. What are they? How are they different from traditional bibliometrics? What are the benefits and disadvantages of them compared to citations? Why have you chosen to use the Altmetric Attention Score and not another altmetric from another data provider? Could the fact that you have only looked at the AAS associated with the top 100 highest-cited papers have biased your analysis? A lot of dubious "research" was circulating on social media during the pandemic. Perhaps there are articles with very few citations that have much bigger AAS scores than the top-cited papers you have looked at? Finally, the term "altmetric" is conflated with "social media" (see lines 422-423, 426, and 492-493), which ignores the fact that the AAS is a composite of mentions on social media AND mainstream news media, policy documents, patents, and other content that cannot be described as "social media."

Conclusions:

This is the weakest part of the paper, as it largely just reiterates the results section and describes the results. The conclusion that the altmetrics analysis shows that "both the academic community and social media [see my comments above] are closely interested in COVID-19 vaccine research" is a particularly shallow one to derive from such a rich dataset. A correlation analysis of altmetrics scores vs. bibliometrics (citations) might have revealed something more interesting but this has not been done.

Minor points:

107: "from each retrieved article/review" - On line 102 it states that reviews were excluded, so are they excluded or not?

Table 1: The United Kingdom is composed of England, Scotland, Wales, and Northern Ireland. I am not quite sure why ISI (now Clarivate) chose to split up the United Kingdom into its separate regions when they started their database. For some reason they haven't done this for other, much larger countries, and it always put the UK at a disadvantage in international comparisons. If publications from the four regions of the UK are combined, does the UK come in second place in this table? The difference between England and China is not that large...

Figure 2 is never explicitly referred to in the text.

309-320: I'm not sure why this section is here. It does not seem to be relevant to the study and could be deleted.

373-375: "In order... local culture." This very strong statement does not seem to be justified by anything in this study.

383-403: Again, I'm not sure why this is here. It is not relevant to the study, which is a bibliometric study.

Author Response

Dear Dr. Reviewer

Thank you very much for your careful review and suggestions with regard to this manuscript. We have studied comments carefully and tried our best to revise and improve the manuscript and made some changes in the manuscript according to your suggestions with tracked changes to highlight the revisions.

This article presents a bibliometric analysis of articles published in the English language on the subject of COVID-19 vaccination. Since I do not have a medical or clinical background, I will limit my comments to the bibliometric content of the article. I assume that at least one of the other reviewers has a background in vaccination and is capable of reviewing those portions of the paper. If this is not the case then the paper should not proceed to publication until this has been rectified.

The authors have taken great pains to acquire a large dataset and they outline their data collection processes thoroughly, allowing others to replicate the data collection, but the analysis is rather superficial. The methodological sections focussing on the analysis require much more detail, and the conclusions are largely just descriptive of the dataset and lack depth.

Answer: Thank you very much for your careful review and valuable suggestions. According to your suggestion, we have tried our best to revise the detail of the manuscript, including Abstract, Methodological sections (data analysis), Results' section (Main publication of countries, institutions, authors and journals, Visualization map of co-authorship countries and institutions, Visualization map of active authors and journals, Visualization map of co-occurrence keywords and burst keywords), Discussion' section and Conclusions' section in the manuscript. Please see page 1-19 of the revised manuscript, which were changed in the manuscript with tracked changes to highlight the revisions (red color).

Data analysis:

There is very little detail about some crucial methodological points. What is "centrality"? What are "citation bursts"?  Simply referring to the article that introduces the CiteSpace software is not enough to make these concepts meaningful for the readership of Vaccines, who will largely be unfamiliar with bibliometrics. The concepts of "link strength" and "silhouette value" which occur regularly in the 'Results' section are never explained.

Answer: Thank you very much for your suggestions. According to your suggestion, we have revised the detail in the methodological sections of manuscript, including centrality, citation bursts, silhouette, modularity, link strength, total link strength and references. Please see page 4, line116 in the revised manuscript.

All inclusion documents were analyzed using CiteSpace software (v.5.8.R3, Phila-delphia, PA, USA) and VOSviewer (v.1.6.17, Leiden, The Netherlands).

CiteSpace software, created by Professor Chaomei Chen [15], was used to analyze information about co-authorship countries, institutions, co-cited references clusters, and keywords of strongest citation bursts. The centrality scores and keywords of cita-tion bursts, heterogeneous networks are detected using CiteSpace software, which can help to timely visualize the research status, hot spots, and frontiers [16].

The centrality indicator for the nodes of the co-citing network is betweenness centrality. The indicator of centrality is used to assess the literature importance and the level of each literature within a co-citation network partially [17]. In the CiteSpace, the normalized value of centrality is between 0 and 1 and the nodes with high centrality scores are important document in a co-cited network [17]. For the keywords of citation bursts, the burst detection algorithm can be suitable to measure the sharp increasing of interest within a short time span [18]. According to citation burst terms extracted from bibliographic records, a present research hot spot and frontier is distinguished in CiteSpace software [15,18]. The metrics of mean silhouette (S) and the modularity (Q) used to evaluate the overall structural properties of the network. The silhouette value above 0.7 (S > 0.7) is considering as the efficient and convincing cluster [19]. The value of modularity is more than 0.3 (Q > 0.3), which means that the network is significant [17].  In current CiteSpace, the following parameter settings were selected [20]: time slicing (2019 to 2022), one year per slice; term source (all selected), node type (one at a time), selection criteria (g-index = 25 in each slice), pruning (no) and visualization (cluster view-static and show merged network).

VOSviewer software, developed by van Eck NJ and Waltman L for viewing bibliometric maps [21], was used to analyze the status of scientific collaborations between authors, journals, and keywords co-occurrence on the vaccine research of COVID-19. Two standard weight attributes are applied which are defined as “links attribute” and “total link strength (TLS) attribute” [22]. For a given item, the links and TLS attributes indicate, respectively, the number of links of an item with other items and the total strength of the links of an item with other items [23]. For example, in the case of co-authorship links between researchers, the TLS attribute shows the total strength of the co-authorship links of a given researcher with other researchers. The following parameter settings of create map wizard were selected [23]: type of analysis (co-authorship), counting method (full counting), unit of analysis (authors), minimum number of documents of author (4) and others parameters (default values) for the authors co-authorship network visualization map; type of analysis (co-citation), counting method (full counting), unit of analysis (cite source), minimum number of citations of source (22) and other parameters (default values) for the the journals co-citations net-work visualization map.

SPSS (IBM SPSS Statistics 19, Inc., Chicago, IL, USA) software was used to analyze Pearson's correlation between total articles output, citations and total link strength of publications. Correlations were considered to be significant when p value < 0.05. 

  1. Chen, C. CiteSpace II: Detecting and visualizing emerging trends and transient patterns in scientifc literature. J. Am. Soc. Inf. Sci. Technol. 2006, 57,359–377.
  2. Tu, S. J., Jin, C., Chen, B. T., Xu, A. Y., Luo, C., & Wang, X. H. Study on the fusion of sports and medicine in China from 2012 to 2021: A bibliometric analysis via CiteSpace. Front. Public Health 2022,10, 939557.
  3. Tao, S., Yang, D., Zhang, L., Yu, L., Wang, Z., Li, L., et al. Knowledge domain and emerging trends in diabetic cardiomyo-pathy: A scientometric review based on CiteSpace analysis. Front. Cardiovasc. Med. 2022, 9, 891428.
  4. Li M.N., Porter A.L., Wang Z.L. Evolutionary trend analysis of nanogenerator research based on a novel perspective of phased bibliographic coupling. Nano Energy 2017, 34,93–102.
  5. Zhong D, Li Y, Huang Y, Hong X, Li J, Jin R. Molecular mechanisms of exercise on cancer: a bibliometrics study and visu-alization analysis via CiteSpace. Front. Mol. Biosci. 2021, 8,797902.
  6. Ma, X.; Zhang, L.; Wang, J.; Luo, Y. Knowledge domain and emerging trends on echinococcosis research: a scientometric analysis. Int. J. Environ. Res. Public Health 2019, 16(5),842.
  7. Van Eck, N. J., Waltman, L. Software survey: VOSviewer, a computer program for bibliometric mapping. Scientometrics 2010, 84(2), 523–538.
  8. Yu, Y., Li, Y., Zhang, Z., Gu, Z., Zhong, H., Zha, Q., et al. A bibliometric analysis using VOSviewer of publications on COVID-19. Ann. Transl. Med. 2020, 8(13), 816.
  9. Van Eck, N. J., Waltman, L. VOSviewer manual. Available online: https://www.vosviewer.com/getting-started#vosviewer-manual (accessed on 10 August 2022).

Results.

This section describes in some detail the surface characteristics of the dataset, but fails to dig under that surface. Thanks to the severity of the pandemic, the COVID-19 scholarly literature went from being non-existent to totally overwhelming within the space of a few short months. As the authors note, 18,285 articles were published in the relatively short period under discussion here, and - crucially - they were published in 3,499 separate journals. Take a look at Figure 5A and tell me whether this is a sensible way to organise the scholarly literature? No group of researchers, even those in prestigious universities, with large library budgets, in rich countries, can be expected to have access to three and a half thousand medical journals. As the pandemic wore on, several academic publishers rushed to make many of these publications available open access, but only because it was politically expedient for them to do so and avoid the accusation that they are holding the world's accumulated knowledge to ransom. Looking at the data presented here, I see the traditional, subscription-journal-based scholarly publishing system reaching (and exceeding) its breaking point. The authors do not go further than simply describing the characteristics of the data.

Answer: Thank you very much for your careful review and suggestions. In CiteSpace and VOSviewer analysis, the information of author, institution, title, abstract, keywords, journal, and publication date was recorded from each retrieved article. According to your suggestion, we have revised this section. Please see page 5, line196; page 6, line196, line 203; page 7, line229; page 9, line 245, line 276; page 10, line 284 in the revised manuscript.

Similarly, no comparison is made of Table 4 (which shows the most "prolific" journals) and Figure 5B (which shows the journals having the most citation impact). Would a more thorough analysis show a negative correlation between "productivity" and impact?

Answer: Answer: Thank you very much for your careful review and suggestions. According to your suggestion, we have revised this section. Please see page 4, line156. Please see page 10, line284.

SPSS (IBM SPSS Statistics 19, Inc., Chicago, IL, USA) software was used to analyze Pearson's correlation between total articles output, citations and total link strength of publications. Correlations were considered to be significant when p value < 0.05.

The results of correlation analysis with Pearson's correlation indicated that a posi-tive correlation was found between total articles output, citations and TLS of journals on the research of COVID-19 vaccine (P<0.05). A significantly strong correlation between the TLS and citations of journals with the correlation coefficient R=0.98 (P<0.01), weak correlation between number of articles and citations of journals (R=0.33 and P<0.01), and a negligible correlation between number of articles and TLS of journals (R=0.27 and P<0.01) were found in the cooperation network on the research of COVID-19 vaccine.

The section on high[ly]-frequent keywords is of very little interest and could be deleted to make room for more methodological detail. The fact that a large number of papers on COVID-19 vaccination use keywords that are all variations on "COVID-19" and "vaccine" is unsurprising. It needs to be better justified if it is to be included.

Answer: Thank you very much for your suggestions. According to your suggestion, we have deleted this section.

The parameters employed in VOSviewer to achieve the clustering shown in Figures 4 and 5 is not made explicit in the 'data analysis' section.

Answer: Thank you very much for your suggestions. According to your suggestion, we have added the parameters of VOSviewer in the methodological sections of manuscript. Please see page 4, line 149 in the revised manuscript.

 The following parameter settings of create map wizard were selected [23]: type of analysis (co-authorship), counting method (full counting), unit of analysis (authors), minimum number of documents of author (4) and others parameters (default values) for the authors co-authorship network visualization map; type of analysis (co-citation), counting method (full counting), unit of analysis (cite source), minimum number of citations of source (22) and other parameters (default values) for the the journals co-citations net-work visualization map.

SPSS (ver. 22) software was used to analyze Pearson's correlation between citations and TLS. Correlations were considered to be significant if the pË‚0.05. 

  1. Van Eck, N. J., Waltman, L. Software survey: VOSviewer, a computer program for bibliometric mapping. Scientometrics 2010, 84(2), 523–538.
  2. Yu, Y., Li, Y., Zhang, Z., Gu, Z., Zhong, H., Zha, Q., et al. A bibliometric analysis using VOSviewer of publications on COVID-19. Ann. Transl. Med. 2020, 8(13), 816.
  3. Van Eck, N. J., Waltman, L. VOSviewer manual. Available online: https://www.vosviewer.com/getting-started#vosviewer-manual (accessed on 10 August 2022).

Altmetrics as a concept are not properly introduced. What are they? How are they different from traditional bibliometrics? What are the benefits and disadvantages of them compared to citations? Why have you chosen to use the Altmetric Attention Score and not another altmetric from another data provider? Could the fact that you have only looked at the AAS associated with the top 100 highest-cited papers have biased your analysis? A lot of dubious "research" was circulating on social media during the pandemic. Perhaps there are articles with very few citations that have much bigger AAS scores than the top-cited papers you have looked at? Finally, the term "altmetric" is conflated with "social media" (see lines 422-423, 426, and 492-493), which ignores the fact that the AAS is a composite of mentions on social media AND mainstream news media, policy documents, patents, and other content that cannot be described as "social media."

Answer: Thank you very much for your careful review and agree with those questions on Altmetric Attention Score. We apologize that we ignore these controversies, and have deleted this section including methodological section, results discussion' section and conclusions' section etc.

Conclusions:

This is the weakest part of the paper, as it largely just reiterates the results section and describes the results. The conclusion that the altmetrics analysis shows that "both the academic community and social media [see my comments above] are closely interested in COVID-19 vaccine research" is a particularly shallow one to derive from such a rich dataset. A correlation analysis of altmetrics scores vs. bibliometrics (citations) might have revealed something more interesting but this has not been done.

Answer: Thank you very much for your suggestions. According to your suggestion, we have revised it in the revised manuscript. Please see page 19, line 586 in the revised manuscript.

A total of 18,285 records in 3,499 journals were retrieved in the Web of Science Core Collection database and included in the final analysis. Over the time number of articles on the vaccine research of COVID-19 has been increased. Although the highest number of publications was from the USA, followed by UK and China, the USA, UK, and Ger-many were the top contributor to the COVID-19 vaccine research field. University of Washington and Harvard Medical School were the leading institutions, while Krammer F from Icahn School of Medicine at Mount Sinai was the most active and influential author to the topic. The New England Journal of Medicine with the highest citation and TLS was the most cited and influential journal in the field of COVID-19 vaccine research. The vaccine research hotspots of COVID-19 focused on the population attitudes towards vaccination, immune-related information analysis of spike proteins, the effectiveness and side effects of COVID-19 vaccine, and the public management of epidemic trans-mission. The findings of this study provide a lobal status, research hotspots and potential trends in the field of COVID-19 vaccine research, which may be assist researchers to mastering the knowledge structure, evaluating and guiding the future development trends of global COVID-19 vaccine.

Minor points:

107: "from each retrieved article/review" - On line 102 it states that reviews were excluded, so are they excluded or not?

Answer: Thank you very much for your suggestions. we have revised it in the revised manuscript. Please see page 4, line 114 in the revised manuscript.

The following information was recorded from each retrieved article: author, institution, title, abstract, keywords, journal, and publication date.

Table 1: The United Kingdom is composed of England, Scotland, Wales, and Northern Ireland. I am not quite sure why ISI (now Clarivate) chose to split up the United Kingdom into its separate regions when they started their database. For some reason they haven't done this for other, much larger countries, and it always put the UK at a disadvantage in international comparisons. If publications from the four regions of the UK are combined, does the UK come in second place in this table? The difference between England and China is not that large...

Answer: Thank you very much for your suggestions. According to your suggestion, we have merged the publications data of four regions of England, Scotland, Wales, and Northern Ireland, and revised it in Table 1 in the revised manuscript. Please see page 5-6, line 184 in the revised manuscript.

The most productive country was the USA with 5,711 (31.2%) publication documents, followed by the UK (n = 2,020, 11.05%), China (n = 1,819, 9.95%), the Italy (n = 1,253, 6.85%), the India (n = 1,193, 6.52%). Among the ten productive countries, the top-3 countries with the highest centrality ranking were UK (0.23), Germany (0.18) and USA (0.05).

Figure 2 is never explicitly referred to in the text.

Answer: Thank you very much for your suggestions. We have revised it in the text. Please see page 5, line 168 in the revised manuscript.

309-320: I'm not sure why this section is here. It does not seem to be relevant to the study and could be deleted.

Answer: Thank you very much for your suggestions. According to suggestion, we have deleted the line 309-320 in the revised manuscript.

373-375: "In order... local culture." This very strong statement does not seem to be justified by anything in this study.

Answer: Thank you very much for your suggestions, and we have revised it in the revised manuscript. Please see page 17, line 470 in the revised manuscript.

Thus, the government and public health authorities should take multi-pronged effort and necessary steps to increase vaccination acceptance and positive attitudes to vaccine.

383-403: Again, I'm not sure why this is here. It is not relevant to the study, which is a bibliometric study.

Answer: Thank you very much for your suggestions. According to your suggestion, we have deleted the line 383-403 in the revised manuscript.

We would like to thank you again for taking the time to review our manuscript.

Best wishes.

Reviewer 2 Report

Although the manuscript is apparently descriptive, it presents a logical organization of excellence, it is didactic, and always starts from a macro vision to a more narrowed and specific one, deepening the initially announced objective. It presents a comprehensive set of data and specificity related to the tables:

Table 1-most productive countries on Covid-19 vaccine research;

Table 2-most prolific institutions;

Table 3- most productive countries;

Table 4-most prolific journals;

Table 5-keywords.

Included here are the mappings of countries and institutions in co-authorship, the clusters, as well as all the software used, which is very relevant.

However, more clarity is suggested in the arrangement of Tables 3A and 3B. The arrangement of tables 3A and 3B is not clear: put 3A and 3B and not only A and B, as shown.

It is also suggested that the conclusions be more contextualized, depending on the countries to which they refer, especially in relation to social, political, economic and scientific development..

Author Response

Dear Dr. Reviewer

Thank you very much for your careful review and suggestions with regard to this manuscript. We have studied comments carefully and tried our best to revise and improve the manuscript and made some changes in the manuscript according to your suggestions with tracked changes to highlight the revisions.

Although the manuscript is apparently descriptive, it presents a logical organization of excellence, it is didactic, and always starts from a macro vision to a more narrowed and specific one, deepening the initially announced objective. It presents a comprehensive set of data and specificity related to the tables:Table 1-most productive countries on Covid-19 vaccine research;Table 2-most prolific institutions;Table 3- most productive countries;Table 4-most prolific journals;Table 5-keywords.Included here are the mappings of countries and institutions in co-authorship, the clusters, as well as all the software used, which is very relevant.However, more clarity is suggested in the arrangement of Tables 3A and 3B. The arrangement of tables 3A and 3B is not clear: put 3A and 3B and not only A and B, as shown.

Answer: Thank you very much for your careful review and valuable suggestions. We have revised it in the manuscript with tracked changes to highlight the revisions (red color). Please see page 7, line223 in the revised manuscript.

It is also suggested that the conclusions be more contextualized, depending on the countries to which they refer, especially in relation to social, political, economic and scientific development.

Answer: Thank you very much for your careful review and valuable suggestions. We have revised it in the manuscript. Please see page 19, line586 in the revised manuscript.

We would like to thank you again for taking the time to review our manuscript.

Best wishes.

Reviewer 3 Report

The paper is devoted to the bibliometric analysis of research papers on vaccines and vaccination against the COVID-19 disease. The authors used the WoS database and considered the period of time till July 2022. This study is potentially useful for the researchers in the field and for policy makers. However, the methodology of the work should be evaluated by the specialists in bibliometric analysis.

The authors can take into account the following remarks.

1.       Explain the meaning of centrality in Table 1.

2.       The sentence in lines 192-193 is not clear.

3.       What is the meaning of colors and clusters in Figure 4?

4.       The same for Figure 5. Moreover, the writings of the figure are very small and mostly unreadable.

5.       What is the meaning of silhouette value on line 241?

6.       What is the meaning of citation burst strength on line 455?

Author Response

Dear Dr. Reviewer

Thank you very much for your careful review and suggestions with regard to this manuscript. We have studied comments carefully and tried our best to revise and improve the manuscript and made some changes in the manuscript according to your suggestions with tracked changes to highlight the revisions.

The paper is devoted to the bibliometric analysis of research papers on vaccines and vaccination against the COVID-19 disease. The authors used the WoS database and considered the period of time till July 2022. This study is potentially useful for the researchers in the field and for policy makers. However, the methodology of the work should be evaluated by the specialists in bibliometric analysis.

Answer: Thank you very much for your careful review and valuable suggestion. According to your suggestion, we have revised the methodology in the text. Please see page 4, line116 in the revised manuscript.

All inclusion documents were analyzed using CiteSpace software (v.5.8.R3, Phila-delphia, PA, USA) and VOSviewer (v.1.6.17, Leiden, The Netherlands).

CiteSpace software, created by Professor Chaomei Chen [15], was used to analyze information about co-authorship countries, institutions, co-cited references clusters, and keywords of strongest citation bursts. The centrality scores and keywords of cita-tion bursts, heterogeneous networks are detected using CiteSpace software, which can help to timely visualize the research status, hot spots, and frontiers [16].

The centrality indicator for the nodes of the co-citing network is betweenness centrality. The indicator of centrality is used to assess the literature importance and the level of each literature within a co-citation network partially [17]. In the CiteSpace, the normalized value of centrality is between 0 and 1 and the nodes with high centrality scores are important document in a co-cited network [17]. For the keywords of citation bursts, the burst detection algorithm can be suitable to measure the sharp increasing of interest within a short time span [18]. According to citation burst terms extracted from bibliographic records, a present research hot spot and frontier is distinguished in CiteSpace software [15,18]. The metrics of mean silhouette (S) and the modularity (Q) used to evaluate the overall structural properties of the network. The silhouette value above 0.7 (S > 0.7) is considering as the efficient and convincing cluster [19]. The value of modularity is more than 0.3 (Q > 0.3), which means that the network is significant [17].  In current CiteSpace, the following parameter settings were selected [20]: time slicing (2019 to 2022), one year per slice; term source (all selected), node type (one at a time), selection criteria (g-index = 25 in each slice), pruning (no) and visualization (cluster view-static and show merged network).

VOSviewer software, developed by van Eck NJ and Waltman L for viewing bibliometric maps [21], was used to analyze the status of scientific collaborations between authors, journals, and keywords co-occurrence on the vaccine research of COVID-19. Two standard weight attributes are applied which are defined as “links attribute” and “total link strength (TLS) attribute” [22]. For a given item, the links and TLS attributes indicate, respectively, the number of links of an item with other items and the total strength of the links of an item with other items [23]. For example, in the case of co-authorship links between researchers, the TLS attribute shows the total strength of the co-authorship links of a given researcher with other researchers. The following parameter settings of create map wizard were selected [23]: type of analysis (co-authorship), counting method (full counting), unit of analysis (authors), minimum number of documents of author (4) and others parameters (default values) for the authors co-authorship network visualization map; type of analysis (co-citation), counting method (full counting), unit of analysis (cite source), minimum number of citations of source (22) and other parameters (default values) for the the journals co-citations net-work visualization map.

SPSS (IBM SPSS Statistics 19, Inc., Chicago, IL, USA) software was used to analyze Pearson's correlation between total articles output, citations and total link strength of publications. Correlations were considered to be significant when p value < 0.05.

  1. Chen, C. CiteSpace II: Detecting and visualizing emerging trends and transient patterns in scientifc literature. J. Am. Soc. Inf. Sci. Technol. 2006, 57,359–377.
  2. Tu, S. J., Jin, C., Chen, B. T., Xu, A. Y., Luo, C., & Wang, X. H. Study on the fusion of sports and medicine in China from 2012 to 2021: A bibliometric analysis via CiteSpace. Front. Public Health 2022,10, 939557.
  3. Tao, S., Yang, D., Zhang, L., Yu, L., Wang, Z., Li, L., et al. Knowledge domain and emerging trends in diabetic cardiomyo-pathy: A scientometric review based on CiteSpace analysis. Front. Cardiovasc. Med. 2022, 9, 891428.
  4. Li M.N., Porter A.L., Wang Z.L. Evolutionary trend analysis of nanogenerator research based on a novel perspective of phased bibliographic coupling. Nano Energy 2017, 34,93–102.
  5. Zhong D, Li Y, Huang Y, Hong X, Li J, Jin R. Molecular mechanisms of exercise on cancer: a bibliometrics study and visu-alization analysis via CiteSpace. Front. Mol. Biosci. 2021, 8,797902.
  6. Ma, X.; Zhang, L.; Wang, J.; Luo, Y. Knowledge domain and emerging trends on echinococcosis research: a scientometric analysis. Int. J. Environ. Res. Public Health 2019, 16(5),842.
  7. Van Eck, N. J., Waltman, L. Software survey: VOSviewer, a computer program for bibliometric mapping. Scientometrics 2010, 84(2), 523–538.
  8. Yu, Y., Li, Y., Zhang, Z., Gu, Z., Zhong, H., Zha, Q., et al. A bibliometric analysis using VOSviewer of publications on COVID-19. Ann. Transl. Med. 2020, 8(13), 816.
  9. Van Eck, N. J., Waltman, L. VOSviewer manual. Available online: https://www.vosviewer.com/getting-started#vosviewer-manual (accessed on 10 August 2022).

The authors can take into account the following remarks.

  1. Explain the meaning of centrality in Table 1.

Answer: Thank you very much for your careful review and suggestion. According to your suggestion, we have revised the centrality in the text. Please see page 4, line116 in the revised manuscript.

The centrality indicator for the nodes of the co-citing network is betweenness centrality. The indicator of centrality is used to assess the literature importance and the level of each literature within a co-citation network partially [17]. In the CiteSpace, the normalized value of centrality is between 0 and 1 and the nodes with high centrality scores are important document in a co-cited network [17].

  1. Tao, S., Yang, D., Zhang, L., Yu, L., Wang, Z., Li, L., et al. Knowledge domain and emerging trends in diabetic cardiomyo-pathy: A scientometric review based on CiteSpace analysis. Front. Cardiovasc. Med. 2022, 9, 891428.
  1. The sentence in lines 192-193 is not clear.

Answer: Thank you very much for your careful review and suggestion. According to your suggestion, we have revised the centrality in the text. Please see page 7, line229 in the revised manuscript.

There were 188 nodes and 223 links in the countries network map and 697 nodes and 749 links in the institutions network map with a loose way. The low density of the country-collaboration and institution-collaboration network map suggested that most countries and institutions were fragmented and lacked consistent and extensive cooperation.

  1. What is the meaning of colors and clusters in Figure 4?

Answer: Thank you very much for your careful review and suggestion. According to your suggestion, we have revised the centrality in the text. Please see page 9, line262, line268 in the revised manuscript.

The size of nods, represented the number of documents, is positively correlated with the connection strength of authors, and authors with the same color in the view belong to the same clustered in cooperation network. As showed in Figure 4, eight major clusters of authors can be distinguished in the co-authorship network.

Figure 4. The authors co-authorship network visualization map in COVID-19 vaccine re-search. The authors co-authorship network visualization map was produced with VOSviewer and the top-3300 authors who collaborated on at least four documents are showed in figure. The size of the nodes represented the number of documents, and authors with the same color in the view belong to the same clustered cooperation network.

  1. The same for Figure 5. Moreover, the writings of the figure are very small and mostly unreadable.

Answer: Thank you very much for your careful review and suggestion. Because of the limitation of page size, the enlarged Figure 5 was re-produced and put it in the supplementary data (Appendix 8).

Figure 5. The top-20 active journals and co-citation cited sources visualization map with in COVID-19 vaccine research by VOSviewer analysis. (A) Network visualization map of top-2980 cited sources (co-citations more than twenty-two); (B) The top-20 highest TLS journals. The size of the nodes represented the citations frequency of journals, and journals with the same color in the view belong to the same clusters in cooperation network. 

  1. What is the meaning of silhouette value on line 241?

Answer: Thank you very much for your careful review and suggestion. We have revised it in the text. Please see page 4, line116 in the revised manuscript.

The metrics of mean silhouette (S) and the modularity (Q) used to evaluate the overall structural properties of the network. The silhouette value above 0.7 (S > 0.7) is considering as the efficient and convincing cluster [19]. The value of modularity is more than 0.3 (Q > 0.3), which means that the network is significant [17].

  1. Tao, S., Yang, D., Zhang, L., Yu, L., Wang, Z., Li, L., et al. Knowledge domain and emerging trends in diabetic cardiomyo-pathy: A scientometric review based on CiteSpace analysis. Front. Cardiovasc. Med. 2022, 9, 891428.
  1. What is the meaning of citation burst strength on line 455?

Answer: Thank you very much for your careful review and suggestion. We have revised it in the text. Please see page 4, line116 in the revised manuscript.

For the keywords of citation bursts, the burst detection algorithm can be suitable to measure the sharp increasing of interest within a short time span [18]. According to citation burst terms extracted from bibliographic records, a present research hot spot and frontier is distinguished in CiteSpace software [15,18].

  1. Chen, C. CiteSpace II: Detecting and visualizing emerging trends and transient patterns in scientifc literature. J. Am. Soc. Inf. Sci. Technol. 2006, 57,359–377.
  2. Li M.N., Porter A.L., Wang Z.L. Evolutionary trend analysis of nanogenerator research based on a novel perspective of phased bibliographic coupling. Nano Energy 2017, 34,93–102.

We would like to thank you again for taking the time to review our manuscript.

Best wishes.

Round 2

Reviewer 1 Report

The manuscript is much improved, but I think it would benefit from an English-language check. Here are a few easily rectified mistakes that I spotted, but there are quite a few more that could be corrected with a proof-read:

Line 24: TLS need to defined in the abstract if it is to be used here.

28: "global" not "lobal"

116: Remove "created by Professor Chaomei Chen" - the citation is sufficient

137: Again, I think the citation is enough here.

217: "with a loose way" does not make any sense to me.

241: "nods" should be "nodes"

399: "and the network was reasonably" - Reasonably what? This is not a complete sentence.

490: "global" not "lobal"

Author Response

Dear Dr. Reviewer

Line 24: TLS need to defined in the abstract if it is to be used here.  28: "global" not "lobal".  116: Remove "created by Professor Chaomei Chen" - the citation is sufficient.  137: Again, I think the citation is enough here. 217: "with a loose way" does not make any sense to me. 241: "nods" should be "nodes". 399: "and the network was reasonably" - Reasonably what? This is not a complete sentence. 490: "global" not "lobal"

Answer: Many thanks for your careful review and suggestions. We have revised those errors of word and sentence, deleted nonsense word of "with a loose way" in the manuscript with tracked changes to highlight the revisions (red color). 

We would like to thank you again for taking the time to review our manuscript.

Best wishes.